# Acceleration or Brakes: Which Is Rational for Cell Cycle-Targeting Neuroblastoma Therapy?

**DOI:** 10.3390/biom11050750

**Published:** 2021-05-18

**Authors:** Kiyohiro Ando, Akira Nakagawara

**Affiliations:** 1Research Institute for Clinical Oncology, Saitama Cancer Center, 818 Komuro, Ina, Saitama 362-0806, Japan; 2Saga International Carbon Particle Beam Radiation Cancer Therapy Center, Saga HIMAT Foundation, 3049 Harakoga-Machi, Saga 841-0071, Japan

**Keywords:** neuroblastoma, cell cycle, MYCN, 11q loss, checkpoint, DNA damage response, replication stress, RAS, telomere, inhibitor

## Abstract

Unrestrained proliferation is a common feature of malignant neoplasms. Targeting the cell cycle is a therapeutic strategy to prevent unlimited cell division. Recently developed rationales for these selective inhibitors can be subdivided into two categories with antithetical functionality. One applies a “brake” to the cell cycle to halt cell proliferation, such as with inhibitors of cell cycle kinases. The other “accelerates” the cell cycle to initiate replication/mitotic catastrophe, such as with inhibitors of cell cycle checkpoint kinases. The fate of cell cycle progression or arrest is tightly regulated by the presence of tolerable or excessive DNA damage, respectively. This suggests that there is compatibility between inhibitors of DNA repair kinases, such as PARP inhibitors, and inhibitors of cell cycle checkpoint kinases. In the present review, we explore alterations to the cell cycle that are concomitant with altered DNA damage repair machinery in unfavorable neuroblastomas, with respect to their unique genomic and molecular features. We highlight the vulnerabilities of these alterations that are attributable to the features of each. Based on the assessment, we offer possible therapeutic approaches for personalized medicine, which are seemingly antithetical, but both are promising strategies for targeting the altered cell cycle in unfavorable neuroblastomas.

## 1. Introduction

Neuroblastomas are the most common extracranial solid tumors of early childhood, occurring in ~150–200 children each year in Japan [1]. The International Neuroblastoma Risk Group (INRG) classification system was developed to categorize neuroblastoma cases into three groups, high, intermediate, and low risk, according to the prognostic variables: stage, age at diagnosis, MYCN oncogene status, chromosome 11q status, DNA ploidy, and the histologic category and grade of tumor differentiation based on the International Neuroblastoma Pathology Classification (INPC) [2]. Moreover, the classification system led to the current consensus of optimal treatment strategies [3]. Intensive multi-modal therapy successfully improved the overall prognosis, but the exploration of novel strategies that are beneficial for patients with poor outcomes (e.g., approximately 50% of cases with high risk) remains urgent [2]. Current standard treatment strategies for high-risk neuroblastoma include induction chemotherapy and surgery, consolidation therapy (high-dose chemotherapy with autologous stem-cell rescue and external radiotherapy), and post-consolidation therapy (anti-ganglioside 2 immunotherapy with cytokines and cis-retinoic acid). The identification of several pathogenic alterations associated with high-risk neuroblastomas has improved our understanding of their biology. Targeted drugs against these specific alterations are in the early phases of clinical trials. However, the efficacy of these drugs for treating neuroblastoma remains unclear, particularly in the high-risk category [3].

Clinically approved selective inhibitors targeting unrestrained cell division in various types of cancer might ultimately converge into two reciprocal rationales, both of which evoke cancer cell death by impacting the aberrant cell cycle. The first applies a “brake” to the cell cycle to stop aberrant proliferation and targets a driver oncogene and downstream effectors driving survival signaling in cancer cells. The second “accelerates” the cell cycle to induce genomic catastrophe and targets genetic vulnerabilities by disrupting the genomic integrity of cancer cells. In the present review, we focused on mutually exclusive genomic aberrations that are common to unfavorable neuroblastomas: MYCN amplification and 11q loss. We propose possible therapeutic approaches to target the intrinsic vulnerabilities of each genetic characteristic.

## 2. Putting the “Brakes” on the Cell Cycle

### 2.1. MYCN

MYCN amplification is reliable marker for the high-risk neuroblastoma group and is prevalent in approximately 20% of patients with neuroblastoma. MYCN encodes the transcriptional factor N-MYC (hereafter referred to as MYCN), which belongs to the MYC proto-oncogene family, along with c-MYC (hereafter referred to as MYC) and L-MYC. A number of downstream targets of MYC have been identified, verifying that MYC has multiple functions in the development and progression of tumors. However, this information further complicates our understanding of the essential contribution of the members of the MYC family as driver oncogenes and obfuscates how MYC signaling could be targeted therapeutically. No drugs currently exist that can inhibit MYCN, either directly or indirectly, for the treatment of neuroblastoma, in spite of its prognostic importance [1]. Mice with conditional loss of Mycn in neuronal progenitor cells showed decreased central nervous system size, caused by a disruption to rapid cell expansion [4], indicating that MYCN accelerates cell proliferation. In the following section, we review several approaches that could be efficacious for targeting molecules that directly contribute to MYCN-driven proliferation in neuroblastomas. 

### 2.2. CDK4

Cyclin-dependent kinase 4 (CDK4) is one of the putative transcriptional targets of MYC [5] and is highly expressed in neuroblastomas with MYCN amplification. Increased CDK4 expression is correlated with poor patient outcomes [6]. CDK6 is a highly homologous serine/threonine kinase that has significant redundancy with CDK4. CDK4/6 bind cyclin D, as well as other cyclin-CDK complexes, and promote cell cycle progression. Mechanistically, the cyclin D-CDK4/6 complex phosphorylates retinoblastoma tumor suppressor protein (Rb), thus activating E2F transcription factors and promoting progression from G0/1 into the S phase. In parallel, cyclin D-CDK4/6 complexes sequester p21^CIP1^ and p27^KIP1^, inhibitors of CDK (CKI), thereby enabling the activation of cyclin E-CDK2 complexes, which are directly involved in overriding the G1/S border, at the restriction point [7,8]. In addition, deletion of Cdk2 in a Myc-overexpressing mouse model (Eµ-Myc) resulted in delayed development of B-cell lymphoma [9], suggesting that Myc-driven S phase entry is important to aberrant cell expansion. Considering that increased expression of MYCN is positively associated with CDK4 expression, targeting CDK, which is responsible for the transition from G0/1 into the S phase, could theoretically be beneficial for putting the “brakes” on the aberrant proliferation in patients with MYCN-amplified neuroblastoma (Figure 1). It has been reported that the CDK4/6 inhibitor, palbociclib, consistently induces G1 cell cycle arrest in neuroblastoma cell lines and reduces levels of phosphorylated Rb. Importantly, the NGP cell lines that possessed CDK4 amplification were sensitive to palbociclib, providing predictive importance of this genomic lesion [10]. In addition to palbociclib, ribociclib and abemaciclib, the other CDK4/6 inhibitors, have significantly prolonged the progression-free survival (PFS) of patients with advanced breast cancer [11]. As several clinical trials in other tumor types are ongoing, more information on predictive biomarkers for optimal treatment strategies for neuroblastomas will be made available in due course.

### 2.3. RAS-MAPK Pathway

Human RAS proto-oncogenes were initially identified as potent transforming genes of the Harvey and Kirsten murine sarcoma viruses, H-ras and K-ras, respectively. A third RAS gene was isolated from a human neuroblastoma cell line and was termed N-RAS [12]. In the early 1980s, fundamental knowledge about tumorigenic conversion, known as transformation, was evidently improved. The transformation from rodent fibroblasts cannot occur with transfection of *RAS* oncogenes alone, unless a cooperating oncogene such as *MYC*, the gene for the polyoma large-T antigen, or adenovirus early region 1A (EIA) oncogene is transfected first [13,14]. Vitally, even mutant RAS cloned from tumor tissue requires concurrent MYC overexpression for transformation to proceed, highlighting the importance of cooperation between MYCN amplification and oncogenic RAS mutations or activation of RAS-related pathways in neuroblastomas. These observations indicate that deficiency of p53 and/or Rb tumor suppressor pathways is also essential for the completion of RAS oncogene-mediated transformation in rodent fibroblasts [15,16]. However, it has been argued that human cells are more resistant than their rodent counterparts to transformation [17], suggesting that humans have an additional requirement of a predisposing factor or immortalized mutation, such as an acquired telomere maintenance mechanism [18].

According to recent genome sequencing analysis, RAS mutations in neuroblastomas are estimated to occur at a rate of approximately 2–3%, but with increasing frequency at relapse [19,20,21,22]. Notably, activation of RAS-related pathways could be a more common aberration in neuroblastomas, rather than mutations to the RAS proto-oncogene itself. This is feasible because aberrant activation of any tyrosine kinase receptors (RTKs), which are frequently overexpressed or mutated as driver oncogenes in various tumors, can activate RAS-related pathways upstream. For this reason, mutations in RAS and RAS-related pathways are usually assessed by a panel of genes [19,20] including anaplastic lymphoma kinase (ALK), which is mutated in familial and sporadic neuroblastomas [23,24]. A series of ALK inhibitors (e.g., brigatinib, ceritinib, lorlatinib, and crizotinib; reviewed in [25]) are currently under clinical development. A signaling cascade of mitogen activated protein kinase (MAPK) is indispensable for the transformation of RAS oncogenes (reviewed in [12]). Consistently, MEK inhibitors [21] efficiently decrease tumor growth of mouse xenografts formed by neuroblastoma cell lines possessing a mutation in the RAS-MAPK relevant pathway, implicating a benefit of putting the “brakes” on the aberrant proliferation in patients with neuroblastomas, presumably even with MYCN amplification (Figure 1). The clinical application of MEK inhibitors provides remarkable benefits in advanced and unresectable melanoma patients with concomitant BRAF inhibitor treatment [26,27]. A series of MEK1/2 inhibitors, trametinib, binimetinib, selumetinib, and cobimetinib, have been tested clinically against various type of cancer, but no outcomes have been published to date [28]. NCI-COG Pediatric MATCH (Molecular Analysis for Therapy Choice) is an ongoing phase II study that individualizes treatment by using targeted drugs, including MEK inhibitors, as pediatric precision medicine based on specific mutation status [29].

### 2.4. Mitotic Kinases

Drugs targeting mitotic kinases, especially aurora kinase A (aurora A) and polo-like kinase 1 (PLK1), have been extensively studied as cancer therapies. These mitotic kinases were identified as therapeutic targets because they are highly expressed, and have prognostic potential, in various cancers [30,31]. Notably, PLK1 and MYCN can activate each other reciprocally [32], and aurora A stabilizes MYCN through direct binding [33], indicating that pharmacological inhibition of either aurora A, MLN8054, and MLN8237 [33], or PLK1, BI 2536, BI 6727 (vorasercib), and GSK461364 [34,35], could be beneficial for treating MYCN-amplified neuroblastomas (Figure 1). Although pharmacological inhibition of mitotic progression is supposed to put the “brakes” on the aberrant proliferation of tumors, considering the essential role of these mitotic kinases for the maintenance of genomic stability by coordinating multiple steps during mitosis, it remains unclear what genetic background could be used without causing adverse effects for patients. The selective inhibitors of aurora A, as well as PLK1, are currently being evaluated in a number of clinical trials with various cancers [36], and detailed information on patient stratification is eagerly awaited.

## 3. “Accelerating” the Cell Cycle to Induce Catastrophic Cell Death

### 3.1. Non-MYCN Amplified Neuroblastomas

Schleiermacher et al. (2012) performed genomic analysis and elucidated that 397 of 505 patients without *MYCN* amplification possessed segmental chromosomal alterations. The presence of either a chromosome 1p deletion, and/or a chromosome 11q deletion, and/or a 17q gain was associated with poorer OS according to a multivariate survival analysis, suggesting that these particular mutations are important for patients with *MYCN* non-amplified neuroblastomas [37]. In particular, among 397 patients with segmental chromosomal alterations, the 11q alteration was observed in approximately 50% of patients (197 patients) and was significantly correlated with poor outcomes in terms of 4-year OS according to univariate analysis. Segmental chromosomal alterations are generally caused by unbalanced chromosomal translocations, which in turn are causally related to irregular DNA repair [38]. Importantly, allelic loss at 11q23 was inversely related to MYCN amplification in tumors [39]. Thus, in the present review, we focus on the 11q loss mutation and discuss possible therapeutic strategies in unfavorable neuroblastoma without MYCN amplification. Combining evidence from comprehensive studies identified that there is a common region of deletion from 11q14 to 11q23, which codes for a series of proteins that function in the DNA damage response (DDR): ATM, MRE11, and H2AFX [39,40].

#### 3.1.1. MRE11

MRE11 is a component of the MRE11/RAD50/NBS1 (MRN) complex, which is considered to be a primary sensor of DDR, and subsequently initiates either homologous recombination (HR) repair or non-homologous end-joining (NHEJ) repair. Hypomorphic mutations of the three genes are associated with human disorders categorized as chromosome instability syndromes, involving developmental and/or degenerative disorders of the nervous system [41,42]. MRE11 potentially has exonuclease activity [43], and a nuclease-dead allele leads to embryonic lethality in homozygous mice [44]. MYCN transcriptionally controls the expression of each component of the MRN complex [45] (Figure 2), and MYCN-dependent replication stress during rapid expansion of neuronal progenitor cells requires MRE11 [46]. These findings illustrate why the 11q aberration is rarely seen in tumors with MYCN amplification, because MRN deficiency could be a synthetic lethal interaction with MYCN amplification. In fact, pharmacological inhibition of MRE11 by mirin inhibited tumor growth of MYCN-amplified neuroblastoma xenografts [46]. Therefore, in non-MYCN amplified neuroblastomas, the loss of MRE11 with the accompanying 11q deletion may contribute to poor prognosis as a consequence of their irregular DNA repair.

#### 3.1.2. ATM

ATM (ataxia-telangiectasia mutated) is positionally cloned from the locus of homozygous germline mutations in ataxia-telangiectasia (A-T) patients. A-T is an autosomal recessive disease with a pleiotropic phenotype, including progressive cerebellar degeneration, chromosomal instability, and cancer predisposition, and is categorized as a chromosome instability syndrome [42,47,48]. In contrast to Mre11-deficient mice, Atm-deficient mice are viable and show many A-T phenotypes, including the occurrence of malignant thymic lymphoma [49,50,51]. Consistently, somatic mutations of ATM were observed in hematologic malignancies at higher frequencies than in other types of tumor [52]. Although these observations indicate that ATM deficiency contributes, at least in part, to malignant transformation of hematopoietic cells due to increased chromosome instability, the mechanisms underlying neuroblastoma pathogenesis remain elusive. ATM shares a phosphoinositide 3-kinase-related domain with other DDR sensor kinases. Upon DDR, ATM phosphorylates and activates p53 tumor suppressor as one of the main substrates [53,54] (Figure 2). ATM can take over double strand brake (DSB) signaling from the MRN complex as one of the central transducers. In addition, this signal transduction is executed by collaboration with the mediator proteins NFBD1/MDC1 (nuclear factor with BRCT domains 1/mediator of DNA checkpoint 1), 53BP1 (p53 binding protein 1), and BRCA1 (Breast cancer 1) [55]. Several lines of evidence from genome-wide association studies (GWAS) in neuroblastomas have identified predisposing SNPs in BARD1 (BRCA1-associated RING domain1), which is a potent interacting protein of BRCA1. As the BRCA1-BRAD1 complex has been implicated in initiating DNA end resection and promoting RAD51 loading in HR [56,57], it remains unclear what the direct roles of BRCA1 and the other mediators of DDR are in neuroblastoma pathogenesis. Importantly, BRCA1-deficient tumors are critically dependent on abrogation of p53 function in mice and humans, suggesting that the combination of DDR deficiency and loss-of-function p53 is highly susceptible for tumorigenesis. Considering the evidence that p53 mutations at diagnosis are rare, but abnormalities of the p53/MDM2/p14ARF pathway are common in relapsed neuroblastomas [58,59,60], DDR deficiency along with p53-related pathway abnormalities may contribute to the pathogenesis of unfavorable neuroblastomas, as they do in other tumors. Notably, disruption of ATM activity causes increased sensitivity to poly-ADP ribose polymerase (PARP) [61,62,63], and the effect was further enhanced in the presence of p53 deficiency [64]. These observations were related to the idea of synthetic lethal interactions, which were first demonstrated in breast and ovarian cancers harboring mutated BRCA1/2 that were extremely sensitive to PARP inhibitors [65,66]. Collectively, the presence of the genomic/genetic characteristics of ATM haploinsufficiency or allelic variants, which are a common hallmark of 11q loss, and the loss of functional p53, which is characterized by abnormalities of the p53/MDM2/p14ARF pathway in neuroblastomas, are likely candidates to sensitize PARP inhibitor-induced cytotoxicity by “accelerating” the cell cycle to induce catastrophic cell death (Figure 2). A study of 237 neuroblastoma specimens revealed that 48.4% of samples possessed single-nucleotide variants (SNVs) and/or copy number alterations in DDR-associated genes located in 11q, suggesting that PARP inhibitors could be indicated in these patients [67]. The Children’s Oncology Group are leading phase II pediatric MATCH trial studies on olaparib, a PARP inhibitor, for the treatment of patients with advanced/relapsed/refractory solid tumors, non-Hodgkin’s lymphoma, or histiocytic disorders with defects in DNA damage repair genes (NCT03233204). The estimated primary completion will be in September 2024.

#### 3.1.3. The ATR-CHK1 Pathway

To expand on other potential targets in neuroblastomas with ATM deficiency, increasing evidence suggests that ATM and the Rad3-related (ATR)-checkpoint kinase 1 (CHK1) axis could be an Achilles’ heel for synthetic lethality. ATR is a phosphoinositide 3-kinase related protein kinase (PIKK), similar to ATM, and is one of the major regulators of DDR. Mutations in the ATR gene are rare, because only heterozygous or hypomorphic mutations are viable. ATR responds to DNA lesions containing single-stranded DNA, such as bulky base adducts, crosslinks, and processed DSBs. The function of ATR was initially thought to be dissociable from that of ATM, but pathway crosstalk is required in certain DNA lesions, especially if either pathway is disrupted (reviewed in [68]). CHK1 is a well-known effector kinase that is activated by ATR and executes S-phase arrest or G2/M cell cycle arrest through phosphorylation of the cell division cycle proteins (cdc) 25A or Wee1 and cdc25c, respectively (Figure 2). Although disruption of either ATR or CHK1 results in replication fork “collapse” in the S phase, a number of studies have suggested that both inhibitors could have a cancer-specific killing ability, including in neuroblastomas [69,70]. This could be achieved via “checkpoint abrogation,” which targets the genetic vulnerability of tumors [71]. Mechanistically, cells survey their own DNA insults through the G1/S, S, and G2/M cell cycle checkpoints (also referred to as DNA damage checkpoints) and subsequently conduct DNA repair to sustain genomic integrity. However, tumors bearing deficiencies in p53 or Rb exhibit impairment to the G1/S checkpoint and largely rely on the G2 checkpoint to repair DNA damage. Therefore, G2 checkpoint inhibition in tumor cells by the CHK1 inhibitor is thought to “accelerate” the cell cycle and leads to mitotic catastrophe caused by premature mitosis with lethal levels of genomic instability (Figure 2). To date, several selective inhibitors of ATR, M6620 (VX-970 or berzosertib), M4344 (VX-803), AZD6738 and BAY1895344, and CHK1, prexasertib (LY2606368), GDC-575 (ARRY-575; RG7741) and CCT245737 (SRA737), have been tested in clinical trials, but have not yet been approved [72,73]. For instance, prexasertib, which is a second-generation small-molecule inhibitor of CHK1, has been evaluated as both a single agent and in combination with other targeted agents or cytotoxic chemotherapies in adults and pediatric patients with tumors [74]. Previous studies have indicated that prexasertib is potently antiproliferative in neuroblastoma cell lines and xenograft mouse models [75,76]. A phase I study in pediatric patients with recurrent or refractory solid tumors (NCT02808650) is currently underway.

#### 3.1.4. ATRX

Telomere maintenance is essential for high-risk neuroblastomas and is comparable to MYCN-amplified tumors [19,77]. Elevated telomerase activity with upregulated telomerase reverse transcriptase (TERT) gene expression is causally related to recurrent genomic rearrangements of the TERT locus and/or its transcriptional activation by MYCN amplification. The remaining subsets of the high-risk group exhibit remarkable alternative lengthening of telomeres (ALT), driven by mutations to the α-thalassemia/mental retardation syndrome X-linked (ATRX) gene. This incompatibility between ATRX mutants and MYCN amplification is caused by the accumulation of detrimental replication stress (RS) (Figure 2), and the ATRX mutation might therefore be a candidate target for synthetic lethality by targeting RS as described below [78].

### 3.2. MYCN-Amplified Neuroblastomas

The MYC proto-oncogene is thought to be closely involved in DNA replication via transcriptional and nontranscriptional mechanisms, and its aberrant activation causes RS [79]. RS is causally related to unchecked cell division and is recognized as a hallmark of cancer, implicating a series of oncogenes that persistently produce recurring critical errors that need to be repaired for cell survival [80]. However, the increased expression of oncogenes generally induces cellular senescence or apoptosis, unless a DDR protein such as ATM, checkpoint kinase 2 (CHK2, a downstream effector kinase of ATM), or p53 is also inhibited [81,82]. Therefore, deficiency of the DDR protein family has been recognized as cancer-prone with concomitant expression of oncogenes. In striking contrast, B-cell lymphomas in Eμ-Myc mice were prevented when they were crossed with a hypomorphic Atr mouse strain (Atr-Seckel), implying that the DDR via the Atr-Chk1 pathway instead maintains the proliferation of Myc-driven lymphomas [83]. Additionally, this illustrates the inverse, i.e., that the ATM is tumor-suppressive whereas the ATR is oncogenic. Similarly, PARP1/2 was highly expressed in MYCN-amplified neuroblastomas compared with neuroblastomas with MYCN single copy. Considering the function of poly (ADP-ribose) polymerases of PARP1/2 involved in DNA damage sensing by PARylating histones and DNA repair factors leading to stabilization of protein complexes for repairing DNA lesions, MYCN-amplified neuroblastomas might predominantly require PARP1/2 due to the increase in their RS [84]. Furthermore, mice bearing isogenic tumors transplanted with Eμ-Myc lymphomas exhibited remarkable regression and apoptotic cell death when treated with a CHK1 inhibitor, highlighting its therapeutic potential. Interestingly, transplanted p53-deficient Eμ-Myc lymphomas are also treatable with CHK1 inhibitors, indicating that this treatment could be beneficial regardless of p53 mutation status. In accordance, CHK1 inhibitor-mediated cell death is dependent on caspase-2, but not p53 [85]. Mechanistic understanding of caspase-2-dependent cell death pathways is gradually improving and may provide an insight into newly targeted drugs in neuroblastomas, introduced in the next section.

Regarding concomitant treatment with targeted drugs and CHK1 inhibitors, we previously demonstrated that an inhibitor of ATM or DNA-dependent protein kinase (DNA-PK) potentiates CHK1 inhibitor-mediated growth suppression in MYCN-amplified neuroblastoma cell lines [86]. In line with this, synthetic lethal screening has identified that deficiency of the Fanconi anemia DNA repair pathway is critical for the sensitivity of CHK1 inhibitors [87]. Therefore, achieving critical levels of RS in the tumor is another rationale for the strategy of “accelerating” the cell cycle by targeting the DNA damage checkpoints including CHK1 (targeting the checkpoint to kill cancer cells is reviewed in [88]). However, it remains controversial whether DDR inhibitors could be beneficial for cancer therapy because ATM deficiency can potentially lead to oncogene-induced proliferation, as shown in *in vitro* transformation experiments and also in the cancer predisposition of A-T patients. Clinical evaluation of the ATM inhibitors, M3451 and AZD0156, is eagerly awaited [72] (Figure 2). In addition, pharmacological inhibition of MRN was proposed to be synthetic lethal in tumors with MYCN amplification. Further assessment and development of MRN inhibitors may therefore be warranted for the treatment of MYCN-amplified neuroblastomas. 

### 3.3. Caspase-2

Caspase-2 is an evolutionarily conserved, from *C. elegans* to mammals, member of the mammalian caspase family, and it is structurally classified as an initiator caspase of the apoptosis cascade. Thereafter, its functional complexity has been unveiled to have a non-apoptotic function and has a role in tumorigenesis in various malignancies (reviewed in [89]). The tumor-suppressive function of caspase-2 was reported in a Eμ-Myc mouse model, in which loss of either a single or both alleles of caspase-2 resulted in accelerated lymphoma tumorigenesis [90]. Loss of caspase-2 inconsistently delayed tumorigenesis in a MYCN mouse model of neuroblastoma (TH-MYCN), in which MYCN was constitutively expressed under the control of the rat tyrosine hydroxylase promoter. Moreover, human neuroblastoma samples showed a correlation between low levels of caspase-2 expression and increased patient survival in the subset of MYCN-non-amplified neuroblastomas [91]. These conflicting observations indicated that caspase-2 has a tissue- and context-specific role in tumorigenesis, with concomitant aberration of driver oncogenes or tumor suppressors. Intriguingly, Atm-/-caspase-2-/-mice showed drastically increased tumor development with an increased extensive aneuploidy compared with Atm-/-mice, suggesting that caspase-2 has a protective role against aneuploidy [92]. Considering that diploidy is strongly associated with unfavorable prognosis of neuroblastomas, the increased aneuploidy caused by low levels of caspase-2 expression may be causally related to increased patient survival. Caspase-2 activation is executed in specific high-molecular-weight protein complexes as activation platforms and potentially restricted to its specific localization, such as the cytoplasm, nucleus, or nucleolus, for the access to different substrates (reviewed in [93]). The most well-known activation platform is PIDDosome, in which the scaffold protein p53-induced protein with a death domain (PIDD) and the adaptor protein RIP-associated ICH-1/CAD-3 homologous protein with a death domain (RAIDD) form an oligomeric complex [94]. Caspase-2-mediated cell death in p53 mutant cells, as mentioned in the above section, has been illustrated by facilitating the PIDDosome assembly at the cell nucleolus in response to combined treatment with the CHK1 inhibitor and irradiation, and required the scaffold protein, Nucleophosmin 1 (NPM1) [85,95,96]. NPM1 genes are frequently mutated and rearranged in a number of human hematological malignancies [97]. *In vivo* experiments have further revealed that Npm1 inactivation leads to unrestricted centrosome duplication and genomic instability in the embryonic development of mice [98]. Therefore, this hypomorphic phenotype of Npm1 has considerable overlap with the elevated frequency of aneuploidy in Atm-/-caspase-2-/-mice, indicating a protective role of PIDDosome in an aberrant DNA ploidy. Mechanistically, in the presence of supernumerary centrosomes generated by disrupting cytokinesis or forcing centrosome duplication, PIDDosome-dependent activation of caspase-2 leads to cleavage of MDM2 and subsequently stabilizes p53, which is capable of inducing p21-mediated cell cycle arrest before further aberrant cell division [99]. In summary, essential functions of caspase-2, in not only pro-apoptotic but also non-apoptotic functions (including cell cycle arrest for maintaining genome integrity), have provided some possible links to neuroblastoma tumorigenesis. From this perspective, CHK1 inhibitors capable of activating caspase-2 may be beneficial, at least in part, for non-MYCN amplified neuroblastomas bearing a deficiency in DNA repair pathways, such as neuroblastomas with 11q loss, since loss of caspase-2 delayed tumorigenesis in TH-MYCN mice but accelerated it in Atm-/-caspase-2-/-mice, as described above. Further elucidation of the caspase-2-associated pathway and its substrates may provide a new strategy of molecular targeted therapy in neuroblastomas.

## 4. Conclusions

Although MYCN amplification has been unequivocally characterized as unfavorable in neuroblastomas, to date, there are no directly druggable targets, which remains a particular clinical dilemma. However, growing comprehensive analysis of MYCN-amplified neuroblastomas dovetails with early discoveries of basic cancer research, on both *in vitro* transformation or *in vivo* tumorigenesis, and CDK4 and RAS-MEK pathways have reemerged as candidates to stop the cell cycle via MYCN-driven aberrant proliferation (Figure 3A). INPC classifies peripheral neuroblastic tumors (neuroblastomas, ganglioneuroblastomas, and ganglioneuromas) into unfavorable and favorable histology. The grading of INPC is assessed according to their morphologic features based on tumor subtype and mitosis–karyorrhexis index (MKI) along with patient age at the time of diagnosis. Regarding morphologic features of INPC, the MKI, which has been implicated as highly proliferative, and the undifferentiated tumor subtype are hallmarks of the unfavorable histology and are frequently detected in MYCN-amplified neuroblastomas [100]. Additionally, overexpression of Cyclin D1, a binding partner with CDK4/6, in neuroblasts and low Cyclin D1 expression in ganglioneuroma indicate an involvement of dysregulation of the G1 cell cycle checkpoint in neuroblastic tumor differentiation [6]. The recent advances in single-cell transcriptomic analyses showed that expression of CCND1, which encodes Cyclin D1, was specific in the neuroblast lineage, whereas expression of the cell cycle inhibitor CDKN1C was specific to chromaffin cells, supporting the importance of Cyclin D1 in developmental trajectories of the peripheral neuroblastic tumors [101]. Altogether, although the direct link between the degree of differentiation in these tumors and possible molecular targeting therapies has been less characterized, the dysregulation of the G1 cell cycle checkpoint may implicate beneficial therapeutic effects by putting the “brakes” on the cell cycle. In line with this, both clinical outcomes and patient stratification, according to genetic characteristics in ongoing clinical trials evaluating approved CDK4/6 and MEK inhibitors, have attracted intense interest.

Loss of 11q is another unfavorable characteristic that is basically incompatible with MYCN amplification and exhibits genetic vulnerability, presumably due to MRE11 or ATM deficiency. This contributes to genomic instability, which is key for malignant transformation but is also a favorable characteristic for treatment with either cytotoxic chemotherapies or targeted drugs. In this respect, targeting other DDR proteins could be an effective treatment for these tumors by further accelerating the cell cycle in cells with defective DNA repair machinery and subsequently inducing replication/mitotic catastrophe. Effective combined treatment with an approved PARP inhibitor and other inhibitors of DDR/DNA damage checkpoints proteins, including ATR, CHK1, and DNA-PK, or other combinations, might improve the specific killing of neuroblastomas harboring 11q loss (Figure 3B).

RS, generated by MYCN-driven unchecked cell division, is a double-edged sword for tumors. Cellular dependency on the ATR-CHK1 pathway is identified as a synthetic lethal interaction with oncogene-induced RS, with clinical application expected for MYCN-amplified neuroblastomas. Consistently, either a single treatment of a PARP inhibitor or combined treatment with a CHK1 inhibitor was demonstrated to increase RS or further potentiate the increased RS in neuroblastoma cells with MYCN amplification [84], supporting the compatibility between PARP inhibitors and inhibitors of the ATR-CHK1 pathway for effectively inducing catastrophic cell death. Taken together with the strategy for treating 11q loss neuroblastomas, the identification of genetic aberrations in genes orchestrating the two core pillars of DDR pathways, the HR and NHEJ, might be worth evaluating to develop a combination treatment with these inhibitors (targeting the DDR for the treatment of high-risk neuroblastoma is reviewed in [73]).

Lastly, accumulating prognostic evidence, as well as the predisposition for telomere maintenance mechanism mutations in high-risk neuroblastomas, emphasizes that cellular immortalization is an essential requirement for transformation. Consistently, RAS, MYC, and TP53, a series of genes that are fundamentally related to classical tumor conversion, are important to high-risk neuroblastoma pathogenesis [19], where the fundamental differences between terminally differentiated epithelial cells and the developmental process of sympathoadrenal cells always need to be taken into account. Considering some patient clusters with tumor relapses that additionally acquire p53 and RAS mutations, as well as mechanisms activating telomerase or the ALT, it may be beneficial that the targeted drugs are offered as first-line therapy to avoid inducing untreatable genetic chaos in tumors. Drugs that either “accelerate” or “brake” the unrestrained cell cycle are expected to prevent tumor growth and may be a powerful strategy for treating unfavorable neuroblastomas.

## Figures and Tables

**Figure 1 biomolecules-11-00750-f001:**
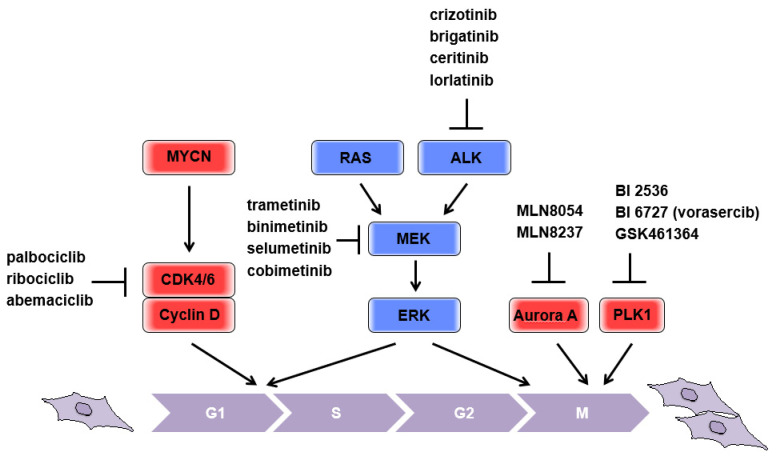
Schematic representation of responsible molecules for applying “brakes” on the neuroblastoma cell cycle. Current druggable targets were depicted.

**Figure 2 biomolecules-11-00750-f002:**
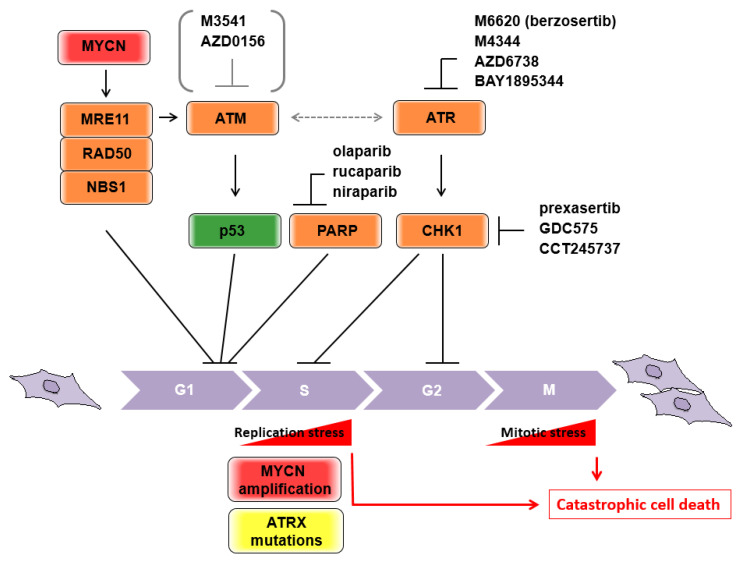
Schematic representation of responsible molecules for the “acceleration” of the neuroblastoma cell cycle. Current druggable targets are depicted. Possible catastrophic cell deaths caused by oncogenic stresses are shown.

**Figure 3 biomolecules-11-00750-f003:**
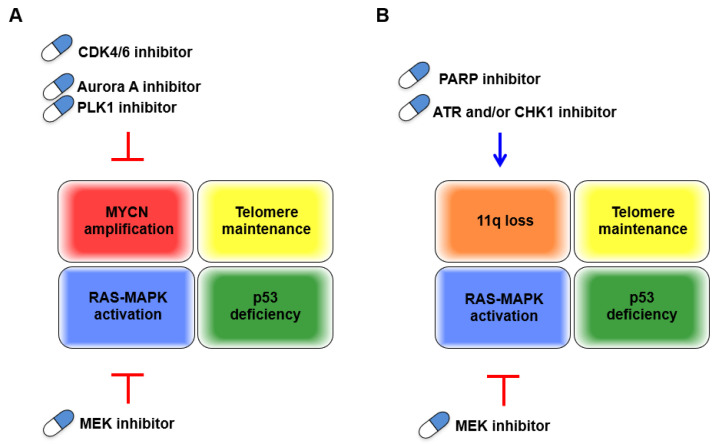
Schematic representation of candidate targeted drugs for two types of unfavorable neuroblastomas. (**A**) MYCN-amplified neuroblastoma, (**B**) non-MYCN amplified neuroblastoma with 11q loss. Red T-bars indicate putting the “brakes” on the cell cycle. Blue arrows indicate “acceleration” of the cell cycle.

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
