# Peer review of "Acceleration or Brakes: Which Is Rational for Cell Cycle-Targeting Neuroblastoma Therapy?"

_biomolecules, 2021, doi:10.3390/biom11050750_

Round 1
Reviewer 1 Report
Targeting the cell cycle is a therapeutic strategy to prevent unlimited cell division.
The present manuscript is well-structured, well-written and easy to understand. A fly in the ointment, when we discuss the cell cycle, we must mention the role of Caspase-2 and the cell fate.
Caspase-2 is the most evolutionarily conserved member of the mammalian caspase family and has been implicated in both apoptotic and non-apoptotic signaling pathways, including tumor suppression, cell cycle regulation, and DNA repair.
Caspase-2 activation platforms; a cytoplasmic platform that is P53-induced protein with a death domain (PIDD) independent, and a nucleolar platform that requires PIDD, each providing access to distinct substrates that regulate cell fate by anti-apoptotic mechanisms.
In the abstract section, the author mentioned as below. This suggests that there is compatibility between inhibitors of DNA repair kinases, such as PARP inhibitors and inhibitors of cell cycle checkpoint kinases. One of the key words of this review is MYCN. We all know, MYCN expression leads to increased replication stress in neuroblastoma cells. This effect is exaggerated by inhibition of PARP, resulting in S-phase specific DNA damage and ultimately increased tumour cell death. High PARP1 and PARP2 expression is significantly associated with high-risk neuroblastoma cases and poor survival, highlighting its previously unrecognized prognostic value for human neuroblastoma. In vitro, PARP1 and 2 are abundant in MYCN amplified and MYCN-overexpressing cells.
In the conclusion section ,the author should discuss a little bit about the PARP inhibitors and MYCN ,and show more perspective on it.
So, I strongly suggest the author add a section of Caspase-2 and cell cycle-targeting neuroblastoma therapy.
The Role of Caspase-2 in Regulating Cell Fate.
Vigneswara V, Ahmed Z.Cells. 2020 May 19;9(5):1259. doi: 10.3390/cells9051259.PMID: 32438737 Free PMC article. Review.
Cell Death Dis. 2014 Aug; 5(8): e1383.PMID: 25144718
An unexpected role for caspase-2 in neuroblastoma
Front. Cell Dev. Biol., 23 December 2020 | https://doi.org/10.3389/fcell.2020.610022
Caspase-2 Substrates: To Apoptosis, Cell Cycle Control, and Beyond
Reviewer 2 Report
Summary:
Ando and Nakagawara review how modulation of cell cycle factors that can either inhibit cell proliferation or induce mitotic catastrophe can be leveraged to target neuroblastomas. This review provides an overview of the major cell cycle checkpoints, how they are connected with cellular DDR signals and the key molecules that regulate these checkpoints. It focuses specifically on how some of these checkpoints are modulated in cancers and identifies the molecules that can be modulated for cancer therapies. The material is concise, balanced and comprehensive with the exception of a few important details outlined below. In summary, the authors have provided an excellent and focused review on cell cycle inhibitors that might be useful for the treatment of neuroblastoma, and hence is suitable for publication in Biomolecules provided they address the minor concerns below, which would strengthen the manuscript further.
Minor comments:
The authors should comment on how selective inhibitors of the cell cycle and DDR pathways might be useful for treating different grades of neuroblastomas. Can they also discuss whether the stage of differentiation of the target neuroblastoma might predispose certain tumor types to a particular cell cycle inhibitor?
Lines 122-124 : The authors should discuss the connection between modulation of CDK4/6 signaling and neuroblastomas as suggested by Rihani et al., Cancer Cell. Int., 2015.
Line 147: The authors should provide a reference supporting the assertion that “[H]uman cells are more resistant that their rodent counterparts to transformation”
Figure 1/2: The white squares in the text after G1, S, G2, M are distracting. This may have been introduced by mistake by a computer program. The authors/copy editor should ensure that these squares are not present in the figure in its final form.
Figure 1: Instead of the blue and white pill indicating mitotic inhibitor, it would be more informative to provide the name or compound number of the chemical used.
